# End-to-End ASR:
# from Supervised to Semi-Supervised Learning with Modern Architectures

Gabriel Synnaeve [*1]    Qiantong Xu [*1]    Jacob Kahn [*1]    Tatiana Likhomanenko [*1]    Edouard Grave [*1]
Vineel Pratap [1]    Anuroop Sriram [1]    Vitaliy Liptchinsky [1]    Ronan Collobert [*1]

## Abstract

We study pseudo-labeling for the semi-supervised training of ResNet, Time-Depth Separable ConvNets, and Transformers for speech recognition, with either CTC or Seq2Seq loss functions. We perform experiments on the standard LIBRISPEECH dataset, and leverage additional unlabeled data from LIBRIVOX through pseudo-labeling. We show that while Transformer-based acoustic models have superior performance with the supervised dataset alone, semi-supervision improves all models across architectures and loss functions and bridges much of the performance gaps between them. In doing so, we reach a new state-of-the-art for end-to-end acoustic models decoded with an external language model in the standard supervised learning setting, and a new absolute state-of-the-art with semi-supervised training. Finally, we study the effect of leveraging different amounts of unlabeled audio, propose several ways of evaluating the characteristics of unlabeled audio which improve acoustic modeling, and show that acoustic models trained with more audio rely less on external language models.

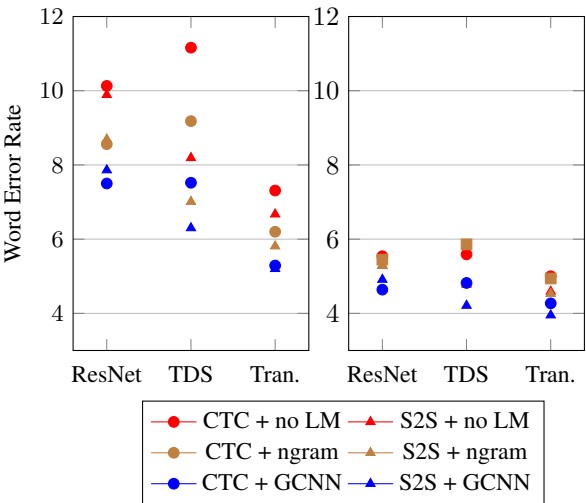

*Figure 1.* WERs on `dev-other` across AM architectures and loss functions. *Left*: WERs of different models trained on LIBRISPEECH with and without beam-search decoding ("no LM" refers to the greedy decoding). Transformer AM architectures outperform others by a large margin. *Right*: WERs of models trained on LIBRIVOX. All models trained on LIBRIVOX significantly outperform their LIBRISPEECH counterparts. The gap between Transformer AMs and other models is much smaller with LIBRIVOX data.

## 1. Introduction

End-to-end speech recognition models are simpler to implement and train than bootstrapped systems. Even given recent promising results from these systems, best-results for common benchmarks are still dominated by classical ASR models; systems requiring force alignment may leave some performance aside for each training step. We set out to study end-to-end systems on LIBRISPEECH (Panayotov et al., 2015) and, without any algorithmic contribution, see if they

can be made to perform as well as more complex training pipelines. The difficulties involved in properly optimizing acoustic models with Connectionist Temporal Classification (CTC) (Graves et al., 2006) or sequence-to-sequence (Seq2Seq) (Sutskever et al., 2014) (v.s. cross-entropy, for instance) combined with more readily-available regularization techniques for classical pipelines make this comparison challenging. Our best acoustic models nonetheless reach 5.17% WER on `test-other`, showing that end-to-end models can compete with traditional pipelines.

As in other domains, self and semi-supervised learning in ASR, where a pretrained network generates and trains on its own labels, yields improvements (Veselý et al., 2017). In end-to-end ASR, pseudo-labeling and self-training can be quite effective, and its effectiveness is further improved

---

[*]Equal contribution  [1]Facebook AI Research, Menlo Park & New York, US, and Paris, France. Correspondence to: Gabriel Synnaeve <gab@fb.com>.

*Published at the workshop on Self-supervision in Audio and Speech at the $37^{th}$ International Conference on Machine Learning*, Vienna, Austria. Copyright 2020 by the author(s).

when more data is available (Kahn et al., 2019a). In this setting, we train a model on LIBRISPEECH, then use that model in conjunction with a language model to generate pseudo-labels from unlabeled audio. We show that with this training scheme, our results *without an external language model* (LM) reach state-of-the-art results that *use an external language model*, with 2.28% and 4.88% Word Error Rate (WER) on `test-clean` and `test-other` respectively. With LM beam-search decoding and rescoring, we reach 2.09% and 4.11% WER on the test set.

While many advances in end-to-end ASR come as the result of neural architecture search (Prabhavalkar et al., 2017; Zhou et al., 2018; Chiu et al., 2018b), we additionally show that simple semi-supervision via pseudo-labeling significantly bridges the performance gap between a variety of different model architectures and loss functions, as shown in Figure 1. In particular, with enough unlabeled audio, Transformer, ResNet, and depthwise-separable convolution-based acoustic models give similar performance with both CTC and Seq2Seq loss functions, suggesting that new techniques in semi-supervision may facilitate equally-significant gains in ASR performance while being applicable to a multitude of end-to-end setups.

## 2. Models

### 2.1. Acoustic Models

In this section, we present the three families of acoustic models (AMs) studied. All AMs output probability distributions over tokens. In particular, we use a set of 10k word pieces (Schuster & Nakajima, 2012; Kudo & Richardson, 2018) generated from the *SentencePiece* toolkit[1]. The choice to use a fixed set of 10k word pieces is made for the simplicity of the comparative study, not the result of a limitation. Similarly, all AMs take 80-channel log-mel filterbanks as input, with STFTs computed on Hamming windows strided by 10ms. This window size is 25ms for Transformer models and 30ms for TDS and ResNet models. All models are trained end-to-end with either CTC or Seq2Seq loss. Given the huge difference between the amounts of data, we prepare two sets of architectures: one for training only on labeled LIBRISPEECH and one for unlabeled LIBRIVOX.

**ResNet Acoustic Model**   ResNets were first introduced in the domain of computer vision (He et al., 2016) and have since been successfully applied to speech recognition (Xiong et al., 2017; Saon et al., 2017; Li et al., 2019b; Wang et al., 2017). ResNets are composed of several blocks of convolutions (in our case only 1-D convolutions), with skip connections. In particular, our ResNet encoder includes 42 convolutional layers each with a kernel size of 3. The encoder first maps the input to an embedding space of size

---

[1] https://github.com/google/sentencepiece

1024 using a single convolutional layer with stride 2; 12 blocks of three 1-D convolutions each follow. Each of the convolutional layers is followed by ReLU, dropout and LayerNorm (Ba et al., 2016). Both the dropout and the number of hidden units increases with the depth of the network. Specific convolution layers are inserted between ResNet blocks in order to upsample when the hidden representation size increases. Our architecture performs significant pooling with respect to the input (16 frames in total, equating to 160 milliseconds) – in addition to the first strided convolutional layer, 3 max pooling layers (each with stride 2) are distributed across the depth of the network (after blocks 3, 7 and 10). Nearly identical encoder architectures are used in front of CTC and Seq2Seq loss functions; the Seq2Seq encoder has its last bottleneck layer removed and lower dropout in deeper layers. The Seq2Seq self-attention decoder for the ResNet architecture is the same as that used with the TDS convolutional AM described below. To better fit the unlabeled data, we increase the model size by increasing the number of channels in each convolution layer.

**Time-Depth Separable (TDS) Convolution Acoustic Model**   We extend the TDS block (Hannun et al., 2019) (which is composed of one 2-D convolution layer and two fully-connected layers with ReLU, LayerNorm and residual connections in between), by increasing the number of channels in the feature maps spanning the two internal fully-connected layers by a factor $F > 1$, so as to increase model capacity. Following (Hannun et al., 2019), 3 sub-sampling layers, i.e. 1-D convolution layers with stride 2, are adopted to ensure an optimal context size for the encoder. For training with only labeled data, we have three groups of TDS blocks with $F = 3$ after each sub-sampling layers. There are 5, 6, and 10 blocks in each group, containing 10, 14, and 18 channels, respectively. To increase model capacity for unlabeled data, the three groups of TDS blocks, having fewer 4, 5, and 6 blocks and $F = 2$ in each, are equipped with much larger 16, 32, and 48 channels. All convolutions in both TDS and sub-sampling layers have kernel size of $21 \times 1$. Identical encoders are used for CTC and Seq2Seq.

Our Seq2Seq self-attention decoder performs $R$ rounds of attention through the same $N$-layers of RNN-GRU each with a hidden unit size of 512 in conjunction with the same efficient key-value attention as in (Hannun et al., 2019; Vaswani et al., 2017):

$$\mathbf{S}_t^r = \text{SOFTMAX}\left(\frac{1}{\sqrt{d}}\mathbf{K}^\top \mathbf{Q}_t^{r-1}\right)\mathbf{V}, \qquad (1)$$

where $[\mathbf{K}, \mathbf{V}]$ is 512-dimensional encoder activation and $\mathbf{Q}_t^r = g(\mathbf{Q}_{t-1}^r, \mathbf{Q}_t^{r-1}) + \mathbf{S}_t^r$ is the query vector at time $t$ in round $r$, generated by the GRU $g(\cdot)$. The initial $\mathbf{Q}_t^0$ is a 512-dimensional token embedding, and the final $\mathbf{Q}_t^R$ is linearly projected to output classes for token classification. In our experiments, $N$ and $R$ are both set to either 2 or 3

based on validation performance. We use dropout in all TDS blocks and GRUs to prevent overfitting.

**Transformer-Based Acoustic Model**  Our transformer-based acoustic models have a small front-end: 3 (LIBRISPEECH AMs) or 6 (LIBRIVOX AM) layers of 1-D convolutions each of kernel width 3 and respective input and output sizes $(80, D_c)$, $(D_c/2, D_c)$, $[(D_c/2, D_c)$, $(D_c/2, D_c)$, $(D_c/2, D_c)$,$]$ $(D_c/2, D_{tr} \times 2)$, with $D_c = 1024$ or $2048$. Each convolution is followed by a GLU activation function (Dauphin et al., 2017) and are striding by 2 each (for 3 consecutive layers), or every other layer (for 6 layers). The output of the front-end for all models is thus strided by 8 frames (80 ms). After the front-end, each Transformer block has 4 attention heads followed by a feedforward network (FFN) with one hidden layer and a ReLU non-linearity. There are two configurations of Transformer blocks: one 24 layer configuration (only for the LIBRISPEECH CTC AM) with dimension $D_{tr} = 1024$ for the self-attention and 4096 for the FFN, and one 36 layer configuration with dimension $D_{tr} = 768$ for the self-attention and 3072 for the FFN. Specifically, given a sequence of $T$ vectors of dimension $d$, the input is represented by the matrix $\mathbf{H^0} \in \mathbb{R}^{d \times T}$, following exactly (Vaswani et al., 2017):

$$\mathbf{Z}^i = \text{NORM}(\text{SELFATTENTION}(\mathbf{H}^{i-1}) + \mathbf{H}^{i-1}),$$
$$\mathbf{H}^i = \text{NORM}(\text{FFN}(\mathbf{Z}^i) + \mathbf{Z}^i),$$

where $\mathbf{Z}$ is the output of the self-attention layer, with a skip connection, and $\mathbf{H}$ is the output of the FFN layer, with a skip connection. As is standard: our NORM is LayerNorm, and self-attention is defined as in Eq. 1, but with $\mathbf{K} = \mathbf{W}_K \mathbf{H}$, $\mathbf{Q} = \mathbf{W}_Q \mathbf{H}$, and $\mathbf{V} = \mathbf{W}_V \mathbf{H}$. For CTC-trained models, the output of the encoder $\mathbf{H}^{L_e}$ is followed by a linear layer to the output classes. For Seq2Seq models, we have an additional decoder, which is a stack of 6 Transformers with encoding dimension 256 and 4 attention heads. The probability distribution of the transcription is factorized as:

$$p(y_1, ..., y_n) = \prod_{i=1}^{n} p(y_i \mid y_0, ..., y_{i-1}, \mathbf{H}^{L_e}), \quad (2)$$

where $y_0$ is a special symbol indicating the beginning of the transcription. For all layers (encoder and decoder – when present), we use dropout on the self-attention and layer drop (Fan et al., 2019), dropping entire layers at the FFN level.

### 2.2. Language Models

In this section, we present external language models (LMs) used in beam-search decoding. We consider $n$-gram LMs as well as convolutional (Dauphin et al., 2017) (GCNN) and Transformer-based LMs. For $n$-gram and GCNN LMs, we train both word-level and word-piece models, and only a word-level Transformer LM. All word-piece LMs are trained on the set of 10k word pieces as outlined in Section 2.1. This ensures that the set of word pieces is consistent across both of the output distributions of the AMs and the candidates the LM scores during beam-search decoding.

For the word-piece and word-level GCNN models, we use the GCNN-14B architecture from (Dauphin et al., 2017) with embedding size 1024 and dropout 0.1. The word-level Transformer LM has the same architecture as (Baevski & Auli, 2019)'s *Google Billion Words* model; we use 16 attention heads and 20 decoder layers with embedding, input and output dimensions of 1280 and 6144 for the FFN with dropout of 0.1.

## 3. Unlabeled Audio Dataset Preparation

LIBRIVOX[2] is a large collection of freely-available audiobooks. Using tools provided with the LIBRILIGHT dataset (Kahn et al., 2019b), we select 72K hours of read speech from English book listings and run several preprocessing steps. After filtering samples to remove readings of duplicate text and corrupted audio, we remove all audio for which the speaker has overlap with a sample in LIBRISPEECH. We run voice activity detection (VAD) using the wav2letter++ framework (Pratap et al., 2018) on the resulting collection of audio with a CTC model trained on LIBRISPEECH, and segment the result into chunks no greater than 36s; the resulting audio corpus contains 53.8K hours of read speech.

We then generate pseudo-labels for this audio using the recipe described in (Kahn et al., 2019a). To generate the pseudo-labels, we use a Transformer AM trained on LIBRISPEECH with CTC loss that achieves a 6.20% WER on dev-other when decoded with a 4-gram word LM – the same model as is listed in Table 3 in the Appendix. We pseudo-label all audio using this AM and run beam-search decoding with a 4-gram word LM described in Appendix A.

## 4. Decoding

Decoding is designed to select the best transcription by leveraging both the posteriors of an acoustic model (AM) and the perplexity of a language model (LM). We perform one-pass beam-search decoding with a single external LM. Optionally, to further improve performance, we use stronger NN-based LMs to rescore the beam. Details on our beam-search decoder algorithm and rescoring are given in Appendix B.

## 5. Experiments

### 5.1. Technical Details

We use the standard splits for LIBRISPEECH (all the available training data was used for training, and two config-

---

[2]https://librivox.org

urations, *clean* and *other*, for validation and test) and the standard LIBRISPEECH LM corpus for LM training. Models are trained using the wav2letter++ toolkit (Pratap et al., 2018); reproduction steps and pre-trained models are open-sourced[3].

**Acoustic Model Training**  All hyper-parameters including model architecture are cross-validated on `dev-clean` and `dev-other`. Given that we have a large family of models, for simplicity and clarity, we only report hyper-parameters ranges in which we search their best values.

Plain SGD with momentum is used to train ResNet and TDS models, and Adagrad (Duchi et al., 2011) to train Transformers. Models are trained on 64 GPUs each with an overall batch size of 256 for ResNet and TDS and 320 for Transformer. With only LIBRISPEECH, all models converged in under a week; with pseudo-labels from LIBRIVOX, training required 2-3 weeks. The initial learning rate for ResNet models is chosen from [0.05, 0.5] , while for TDS and Transformer models, the range decreases to [0.01, 0.03]. Specifically, for Transformers, we apply a linear learning rate warm-up schedule for either 32k or 64k updates. For fully-supervised training with LIBRISPEECH, the learning rate is halved every 90 epochs for Transformer models, and 150 epochs for ResNet and TDS models. With LIBRIVOX, however, we only halve the learning rate once in the middle of the training. For TDS and ResNet models, we use momentum in the range [0.1, 0.6]. With respect to regularization, we use 0.2 dropout everywhere (front-end, encoder, decoder), and layer drop for all Transformer blocks. Dropout in TDS blocks and ResNet convolutions is in the range [0.05, 0.2] and increases with depth. For Seq2Seq training, we run 3 epochs of attention-window pretraining, and use 99% of teacher forcing (1% of uniform output sampling). We also use 10% dropout in the decoder for TDS (and 0.1 dropout and 0.1 layer drop in the decoder for Transformers), together with 5% label smoothing, 1% random sampling and 1% word piece sampling. All models use SpecAugment (Park et al., 2019) with an LD policy.

**Language Model Training**  All LMs in this section are trained on the standard LIBRISPEECH LM corpus. All word-level LMs use the same vocabulary for training. $n$-gram LMs are trained with the KenLM toolkit (Heafield, 2011), while the GCNN and Transformer LMs are trained with fairseq[4] toolkit (Ott et al., 2019). The word-level 4-gram and GCNN are trained in the same way as (Likhomanenko et al., 2019). We also train a 6-gram word-piece LM, which has a similar context size to a word-level 4-gram LM, and prunes 5-grams appearing once and 6-gram appearing twice or fewer. The word-piece and word-level GCNN models

---

[3]https://github.com/facebookresearch/wav2letter
[4]https://github.com/pytorch/fairseq

are trained with Nesterov accelerated gradient descent (Nesterov, 1983) on 8 GPUs for 22 epochs with a step-wise learning rate schedule starting from 1 and decreasing by a factor of 5 when the loss is on the plateau. Gradient clipping and weight normalization are used following (Dauphin et al., 2017). The word-level Transformer LM is trained with Nesterov accelerated gradient descent on 128 GPUs for 100 epochs with an inverse square root learning rate schedule. During the first 16k iterations, a warm-up schedule that linearly increases the learning rate from 1e-7 to 1 is used. Word-level perplexities of all LM variants are listed in Table 1.

*Table 1.* Word-level perplexities of LMs on LIBRISPEECH. Perplexity is computed without unknown words.

| LANGUAGE MODEL | DEV-CLEAN | DEV-OTHER |
|---|---|---|
| WORD 4-GRAM | 148.0 | 136.6 |
| NO LIBRIVOX OVERLAP | 152.8 | 140.0 |
| WP 6-GRAM | 145.4 | 133.7 |
| WP GCNN (188M) | 61.7 | 61.9 |
| WORD GCNN (319M) | 57.0 | 57.9 |
| WORD TRANSF. (562M) | 48.2 | 50.2 |

### 5.2. Results

**LIBRISPEECH Results**  All our results for LIBRISPEECH are listed in the top of Table 3 in Appendix. We present results under three scenarios: without any decoding nor external LM (greedy decoding), with one-pass decoding only, and with decoding followed by beam rescoring. The decoding beam size is usually 50 and 500 for Seq2Seq and CTC respectively. We use a beam size of 250 for CTC decoding with a GCNN LM. We train strong baselines on simple ResNet architectures and improve the TDS models significantly compared to past results (Hannun et al., 2019). These convolutional models outperform end-to-end biLSTM models from (Lüscher et al., 2019). Our best acoustic models are Transformers-based and reach 6.98% without any decoding on `test-other` and 5.17% with decoding and rescoring, demonstrating that end-to-end training can perform as well as traditional bootstrapped systems.

**LIBRIVOX Results**  Assuming all pseudo-labels are ground-truth, we train acoustic models on a combination of the 960 hours of labeled audio from LIBRISPEECH in conjunction the pseudo-labeled audio from LIBRIVOX, where batches are uniformly sampled (without weighting) from both LIBRISPEECH and LIBRIVOX datasets. Transformer AMs with both CTC and Seq2Seq loss were trained for 5 days on this combined dataset, achieving WERs on `test-other` of 4.88% and 2.28% on `test-clean` without decoding or use of an LM, which is state-of-the-art even amongst pipelines that use an LM.

*Table 2.* WERs on LIBRISPEECH development and test sets. Our best results are shown in the bottom section (with the number of parameters), and are both trained with Seq2Seq loss. Full results can be found in Appendix Table 3.

| AM | | LM | | DEV | | TEST | |
|---|---|---|---|---|---|---|---|
| TYPE | LEXICON | TYPE | LEXICON | CLEAN | OTHER | CLEAN | OTHER |
| LAS (PARK ET AL., 2019) | 16K WP | - | - | | | 2.8 | 6.8 |
| DECODING | 16K WP | RNN | 16K WP | | | 2.5 | 5.8 |
| HMM/BILSTM | 12K CDP | 4GRAM+LSTM | WORD | 2.2 | 5.1 | 2.6 | 5.5 |
| + TRANSF. RESCORING | 12K CDP | + TRANSF. | WORD | 1.9 | 4.5 | 2.3 | 5.0 |
| (LÜSCHER ET AL., 2019) | | | | | | | |
| TRANSFORMERS | BPE | RNN | WORD | 2.2 | 5.6 | 2.6 | 5.7 |
| (KARITA ET AL., 2019) | | | | | | | |
| CONV. TRANSF. | 6K TRIPHONES | 3GRAM, RESCORED | WORD | 1.8 | 5.8 | 2.2 | 5.7 |
| (HAN ET AL., 2019) | | + TDNN + LSTM | | | | | |
| CONV. TRANSF. | CHENONES | 4GRAM | WORD | | | 2.60 | 5.59 |
| + TRANSF. RESCORING | CHENONES | TRANSF. | WORD | | | 2.26 | 4.85 |
| (WANG ET AL., 2019) | | | | | | | |
| TRANSF. (270M) – LIBRISPEECH | 10K WP | - | - | 2.54 | 6.67 | 2.89 | 6.98 |
| + DECODING/RESCORING | 10K WP | GCNN + TRANSF. | WORD | 2.07 | 4.79 | 2.37 | 5.17 |
| TRANSF. (296M) – LIBRIVOX | 10K WP | - | - | 2.12 | 4.59 | 2.28 | 4.88 |
| + DECODING/RESCORING | 10K WP | GCNN + TRANSF. | WORD | 2.00 | 3.65 | **2.09** | **4.11** |

Results with decoding/rescoring are shown in Table 2, where we reach 2.09% and 4.11% on `test-clean` and `test-other`, respectively, and are further improvements on the state-of-the-art. From ablations study, Appendix C and D, we found interesting outcomes: i) increasing the amount of pseudo-labels strictly improves performance, ii) models trained on LIBRIVOX pseudo-labels alone outperform models trained on LIBRISPEECH, iii) a large collection of pseudo-labeled audio helps to learn better acoustic representation and transfer LM knowledge so there is no longer benefit much from decoding with an external LM.

## 6. Related Work

Deep neural networks were reintroduced in ASR with HMMs (Hinton et al., 2012), and many of state-of-the-art models still rely on force alignment (Han et al., 2017; Lüscher et al., 2019; Karita et al., 2019). Nonetheless, there have been increasingly competitive end-to-end results trained with CTC (Graves & Jaitly, 2014; Amodei et al., 2016), ASG (Collobert et al., 2016; Zeghidour et al., 2018), LF-MMI (Hadian et al., 2018), sequence-to-sequence (Chan et al., 2016; Chiu et al., 2018a), transduction (Prabhavalkar et al., 2017; He et al., 2019), and differentiable decoding (Collobert et al., 2019a). *Listen Attend and Spell* (Chan et al., 2016) is a family of end-to-end models based on biL-STMs which achieved state-of-the-art results with improved regularization through data augmentation (Park et al., 2019); we consequently use SpecAugment in all of our experiments. Seq2Seq models are not limited to RNNs; time-depth separable convolutions also give strong results (Hannun et al., 2019). Our best models are transformer-based, as (Lüscher et al., 2019; Karita et al., 2019), which give good results

in Seq2Seq settings even without external LMs (Mohamed et al., 2019). In ASR, semi-supervised pseudo-label-style self-training has been explored generally in end-to-end settings in (Soltau et al., 2016; Li et al., 2019a; Kahn et al., 2019a) for both low-resource (Veselỳ et al., 2017; Cui et al., 2017) and large-scale (Parthasarathi & Strom, 2019) setups.

## 7. Discussion

We presented state-of-the-art results on LIBRISPEECH with end-to-end methods. While allowing for lexicon-free decoding, the 10k word-piece tokens used during training limit the amount of striding we can use in our model architectures and can be replaced by AMs outputting words with an arbitrary lexicon (Collobert et al., 2019b). As relative WER gains due to language models shrink (from ≈20% relative-WER without LIBRIVOX to ≈10% with, for GCNN decoding), and as we showed that AMs learn LM-level information, differentiable decoding (Collobert et al., 2019a) is a possible avenue for single-stage AM + LM joint training.

We show the effectiveness of a simple pipeline that does not require many training steps. In light of our semi-supervised results without decoding or an LM, we think Seq2Seq/CTC losses, transducers, and differentiable decoding are viable methods to achieve end-to-end state-of-the-art results, without external LMs, through semi-supervised learning.

## 8. Acknowledgements

We would like to thank Steven Garan for audio recordings of shuffled sentences from LIBRISPEECH `dev-other`.

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

*Table 3.* Word error rates on LIBRISPEECH's development and test sets. Our models listed in the top and bottom blocks are trained with CTC and Seq2seq losses respectively.

| AM | | LM | | DEV | | TEST | |
|---|---|---|---|---|---|---|---|
| TYPE | LEXICON | TYPE | LEXICON | CLEAN | OTHER | CLEAN | OTHER |
| **CTC** | | | | | | | |
| RESNET (306M) | 10K WP | - | - | 3.93 | 10.13 | 4.08 | 10.03 |
| DECODING | | ZEROLM | LEX | 3.76 | 9.7 | 4.07 | 9.77 |
| DECODING | | 4GRAM | WORD | 3.29 | 8.56 | 3.68 | 8.69 |
| DECODING | | GCNN | WORD | 2.99 | 7.50 | 3.28 | 7.53 |
| RESNET (500M) LIBRIVOX | 10K WP | - | - | 2.34 | 5.54 | 2.55 | 5.99 |
| DECODING | | ZEROLM | LEX | 2.37 | 5.45 | 2.73 | 5.96 |
| DECODING | | 4GRAM | WORD | 2.34 | 5.23 | 2.68 | 5.75 |
| DECODING | | GCNN | WORD | 2.19 | 4.64 | 2.45 | 5.13 |
| TDS (200M) | 10K WP | - | - | 4.22 | 11.16 | 4.63 | 11.16 |
| DECODING | | ZEROLM | LEX | 3.93 | 10.61 | 4.44 | 10.67 |
| DECODING | | 4GRAM | WORD | 3.49 | 9.18 | 3.98 | 9.53 |
| DECODING | | GCNN | WORD | 2.92 | 7.52 | 3.40 | 8.05 |
| TDS (500M) LIBRIVOX | 10K WP | - | - | 2.44 | 5.70 | 2.66 | 6.11 |
| DECODING | | ZEROLM | LEX | 2.47 | 5.61 | 2.86 | 6.18 |
| DECODING | | 4GRAM | WORD | 2.44 | 5.33 | 2.81 | 5.91 |
| DECODING | | GCNN | WORD | 2.26 | 4.71 | 2.55 | 5.24 |
| TRANSF. (322M) | 10K WP | - | - | 2.99 | 7.31 | 3.09 | 7.40 |
| DECODING | | ZEROLM | LEX | 2.85 | 6.98 | 3.14 | 7.23 |
| DECODING | | 4GRAM | WORD | 2.63 | 6.20 | 2.86 | 6.72 |
| + RESCORING | | GCNN + TRANSF. | WORD | 2.18 | 4.90 | 2.44 | 5.36 |
| DECODING | | GCNN | WORD | 2.35 | 5.29 | 2.57 | 5.85 |
| + RESCORING | | GCNN + TRANSF. | WORD | 2.20 | 4.94 | 2.47 | 5.45 |
| TRANSF. (299M) LIBRIVOX | 10K WP | - | - | 2.28 | 5.00 | 2.39 | 5.35 |
| DECODING | | ZEROLM | LEX | 2.31 | 4.94 | 2.58 | 5.42 |
| DECODING | | 4GRAM | WORD | 2.24 | 4.59 | 2.52 | 5.22 |
| + RESCORING | | GCNN + TRANSF. | WORD | 1.99 | 3.91 | 2.28 | 4.50 |
| DECODING | | GCNN | WORD | 2.09 | 4.27 | 2.41 | 4.79 |
| + RESCORING | | GCNN + TRANSF. | WORD | 2.01 | 3.95 | 2.31 | 4.54 |
| **SEQ2SEQ** | | | | | | | |
| RESNET (389M) | 10K WP | - | - | 3.51 | 9.89 | 4.92 | 10.33 |
| DECODING | | ZEROLM | LEXFREE | 3.42 | 9.60 | 4.31 | 9.59 |
| DECODING | | 6GRAM | 10K WP | 3.05 | 8.69 | 3.88 | 8.88 |
| DECODING | | GCNN | 10K WP | 2.78 | 7.86 | 3.79 | 8.21 |
| RESNET (500M) LIBRIVOX | 10K WP | - | - | 2.27 | 5.29 | 2.86 | 5.88 |
| DECODING | | ZEROLM | LEXFREE | 2.26 | 5.28 | 2.67 | 5.54 |
| DECODING | | 6GRAM | 10K WP | 2.29 | 5.25 | 2.69 | 5.62 |
| DECODING | | GCNN | 10K WP | 2.26 | 4.91 | 2.66 | 5.31 |
| TDS (190M) | 10K WP | - | - | 3.20 | 8.20 | 3.43 | 8.30 |
| DECODING | | ZEROLM | LEXFREE | 2.89 | 8.00 | 3.24 | 7.99 |
| DECODING | | 6GRAM | 10K WP | 2.76 | 7.01 | 3.18 | 7.16 |
| DECODING | | GCNN | 10K WP | 2.54 | 6.30 | 2.93 | 6.43 |
| TDS (500M) LIBRIVOX | 10K WP | - | - | 2.17 | 4.78 | 2.37 | 5.15 |
| DECODING | | ZEROLM | LEXFREE | 2.20 | 4.80 | 2.38 | 5.11 |
| DECODING | | 6GRAM | 10K WP | 2.18 | 4.61 | 2.35 | 5.02 |
| DECODING | | GCNN | 10K WP | 2.08 | 4.21 | 2.24 | 4.61 |
| TRANSF. (270M) | 10K WP | - | - | 2.54 | 6.67 | 2.89 | 6.98 |
| DECODING | | ZEROLM | LEXFREE | 2.49 | 6.32 | 2.75 | 6.58 |
| DECODING | | 6GRAM | 10K WP | 2.29 | 5.81 | 2.72 | 6.23 |
| + RESCORING | | GCNN + TRANSF. | WORD | 2.13 | 5.00 | 2.51 | 5.47 |
| DECODING | | GCNN | 10K WP | 2.12 | 5.20 | 2.40 | 5.70 |
| + RESCORING | | GCNN + TRANSF. | WORD | 2.10 | 4.79 | 2.33 | 5.17 |
| TRANSF. (296M) LIBRIVOX | 10K WP | - | - | 2.12 | 4.59 | 2.28 | 4.88 |
| DECODING | | ZEROLM | LEXFREE | 2.10 | 4.53 | 2.27 | 4.80 |
| DECODING | | 6GRAM | 10K WP | 2.06 | 4.32 | 2.25 | 4.70 |
| + RESCORING | | GCNN + TRANSF. | WORD | 1.91 | 3.76 | 2.10 | 4.20 |
| DECODING | | GCNN | 10K WP | 1.97 | 3.95 | 2.17 | 4.37 |
| + RESCORING | | GCNN + TRANSF. | WORD | 2.00 | 3.65 | 2.09 | 4.11 |

## A. Pseudo-Labeling: Text Corpus Preparation and $n$-gram LM Training

The LIBRISPEECH language model corpus[5] contains text from 14500 public domain books taken from the Gutenberg project[6]. Given that pseudo-labels are generated with a beam-search decoding procedure that integrates a language model, it is important that the corpus used to train the language model does not have overlap with the unlabeled audio, else information about the ground truth labels for that unlabeled audio may be explicitly embedded in the LM. We remove all text from the LIBRISPEECH language model training corpus that is ground truth for any of the unlabeled audio from the subset of LIBRIVOX.

To do so, we follow several steps. Firstly, we filter out all books from the LIBRISPEECH LM corpus with IDs present in LIBRIVOX. Secondly, after normalizing all titles (removing punctuation, casing, and non-alphanumeric tokens), we remove all titles with zero Levenshtein distance between titles from the LIBRIVOX and the LIBRISPEECH LM corpuses. We use a Levenshtein metric over words rather than tokens for improved performance. We then find titles with nonzero but low similarity scores isolated via the following conditions. Given two book title strings $s_1$ and $s_2$, and constants $\alpha$ and $\beta$:

$$\max\{|s_1|, |s_2|\} - \min\{|s_1|, |s_2|\} < \alpha \cdot \min\{|s_1|, |s_2|\} \ \&$$

$$\text{Levenshtein}(s_1, s_2) \le \beta \cdot \max\{|s_1|, |s_2|\}$$

where notation $|s|$ refers to the number of words in the string $|s|$, and 0.75 and 0.3 were used as values for $\alpha$ and $\beta$, respectively. These constants are found empirically to remove obviously different titles and to have reasonable number of pairs ( 10k) for further manual check. Titles that are manually matched are removed to create the final corpus; 13% of the original LIBRISPEECH-LM corpus was filtered with the aforementioned steps.

Before training LMs, we normalize the filtered corpus so as to mimic the original normalization procedure found in LIBRISPEECH. 88% of our normalized/filtered corpus has identical normalized text compared to the original LIBRISPEECH LM corpus. As a result of our using a different tokenizer, sentence boundaries may differ across corpuses, as may abbreviations (e.g. we map '&c' to 'et cetera').

A 4-gram language model is trained with the resulting corpus using the KenLM toolkit (Heafield, 2011) and the top 200k words as vocabulary. The model is trained without pruning (183k of the top 200k words are the same as the original LIBRISPEECH LM corpus). This model is then used at beam-search decoding time in conjunction with an acoustic

[5] http://www.openslr.org/11/
[6] https://www.gutenberg.org/

model trained on LIBRISPEECH to generate pseudo-labels on the subset of LIBRIVOX detailed in Section 3. During beam-search decoding we use a lexicon which is constructed from the LIBRISPEECH train sets only.

The perplexity difference between the 4-gram LM trained on the filtered corpus and the 4-gram LM trained on original LIBRISPEECH LM corpus is small. The word perplexity of each model is shown in Table 1. Beam-search decoding of a Transformer AM trained on LIBRISPEECH with an LM trained on the filtered corpus results in only a 0.05% absolute WER increase on dev-other compared to decoding with an $n$-gram trained on the full corpus.

## B. Decoding

### B.1. Beam-search Decoder

In our experiments, we use lexicon-based and lexicon-free beam-search decoders following (Collobert et al., 2016; Likhomanenko et al., 2019) with either $n$-gram or GCNN LMs. The lexicon-based decoder, whose search space is limited to the words in the lexicon, is used for CTC models with a word-level LM. The lexicon-free decoder is capable of generating words with arbitrary spelling and is used for S2S models with a word-piece LM. The decoder takes as input posteriors from an acoustic model, a prefix trie built on a lexicon, and an external LM. We tune the language model weight $\alpha$ and the word insertion penalty $\beta$ on validation sets (dev-clean and dev-other). The decoder outputs a transcription $\hat{\mathbf{y}}$ that maximizes

$$\log P_{AM}(\hat{\mathbf{y}}|\mathbf{x}) + \alpha \log P_{LM}(\hat{\mathbf{y}}) + \beta|\hat{\mathbf{y}}|.$$

To stabilize the Seq2Seq beam search, we introduce an EOS-penalty $\gamma$ to hypothesis that have finished in an end-of-sentence token. $\gamma$ is tuned together with other hyperparameters and our experiments show that this strategy effectively prevents the decoder from early-stopping. To improve decoding efficiency, we also incorporate the thresholding technique in (Hannun et al., 2019) and strategies mentioned in (Zeghidour et al., 2018) including hypothesis merging, score caching, and batched LM forwarding. For CTC decoding, following (Park et al., 2018), only the blank token is considered if its posterior probability is greater than 0.95.

### B.2. Rescoring

After acquiring the transcriptions of the $N$-best hypotheses from the one-pass beam-search decoder, we use an external word-level GCNN LM and a Transformer LM to evaluate their log-probabilities, denoted as $\log P_1(\hat{\mathbf{y}})$ and $\log P_2(\hat{\mathbf{y}})$ respectively. We then perform rescoring to reorder the hypotheses according to the following score:

$$\log P_{AM}(\hat{\mathbf{y}}|\mathbf{x}) + \alpha_1 \log P_1(\hat{\mathbf{y}}) + \alpha_2 \log P_2(\hat{\mathbf{y}}) + \beta|\hat{\mathbf{y}}|,$$

where $\alpha_1$, $\alpha_2$, $\beta$ are hyper-parameters of the rescoring algorithm optimized on the validation set and $|\hat{\mathbf{y}}|$ is the transcription length in characters (including the spaces between words). In order to diversify the hypotheses in the beam, to increase the probability that the correct transcription is included, we usually relax the threshold in the decoder and increase beam size when dumping beam candidates.

## C. Ablations

### C.1. Varying the amount of unlabeled audio

In this study, we train on several different randomly-selected subsets of pseudo-labels from the original collection generated as described in Section 3. Results are given in Table 4. Increasing the amount of pseudo-labels strictly improves performance. The listed 53.8k hour result is using the fully-prepared dataset as outlined in Section 3. WERs given are without decoding after 800k iterations of training.

*Table 4.* WERs of a Transformer AM architecture outlined in section 2.1 trained with Seq2Seq loss on LIBRISPEECH with different amounts of pseudo-labeled audio from LIBRIVOX.

| TRAINING DATASET (HOURS) | DEV-CLEAN | DEV-OTHER |
|---|---|---|
| LS ONLY | 2.54 | 6.67 |
| LS + 1K LV | 2.35 | 5.56 |
| LS + 3K LV | 2.21 | 5.16 |
| LS + 10K LV | 2.11 | 4.95 |
| LS + 53.8K LV | 2.11 | 4.59 |

*Table 5.* WERs of a Transformer AM when trained with pseudo-labels generated with a decoder integrating an LM that contains overlapping text with unlabeled audio versus an LM with no overlap. Results are shown after decoding with the word 4-gram language model described in Section 2.2.

| MODEL | OVERLAP | DEV-OTHER | TEST-OTHER |
|---|---|---|---|
| TRANS. S2S | NO | 4.58 | 4.90 |
| | YES | 4.51 | 4.87 |
| TRANS. CTC | NO | 4.92 | 5.47 |
| | YES | 4.80 | 5.33 |

### C.2. Generating pseudo-labels with an LM containing overlapping text

As discussed in Appendix A, using an LM to generate pseudo-labels that was trained with a corpus that includes ground truth text from unlabeled audio introduces an overlap that may unrealistically improve the quality of pseudo-labels. We show that the effect of using an LM trained with a small amount of overlapping text to generate pseudo-labels has

only a small effect on the performance of models trained on those pseudo-labels.

Table 5 contains results for Transformer AMs with both CTC and Seq2Seq loss as described in 2.1 trained on pseudo-labels generated with a decoding step that uses an LM trained on an overlapping versus non-overlapping corpus. The models used are of the same architecture as described in Section 2.1. There is a small improvement in dev-other performance for pseudo-labels generated from an overlapping LM, but both models generalize very similarly.

### C.3. Training on pseudo-labels only

Models trained on LIBRIVOX pseudo-labels alone outperform models trained on LIBRISPEECH. As outlined in Section 5, all acoustic models are trained on a combination of LIBRISPEECH and pseudo-labeled LIBRIVOX audio. In this setup, it is difficult to disambiguate the importance of the pseudo-labeled audio compared to supervised data from LIBRISPEECH. To test the quality of pseudo-labels in isolation, we trained a CTC-based Transformer model similar to that described in Section 2.1 to compare directly with the CTC-based transformer AM used to generate the pseudo-labels described in Section 3. We compare the resulting AM-only performance on the LIBRISPEECH development sets. Without decoding, the resulting LIBRIVOX pseudo-label-only model achieves WERs of 2.38% and 5.43% on dev-clean and dev-other respectively, which improves over the LIBRISPEECH-only baseline's 2.99% and 7.31%, respectively. The volume, quality, and diversity of the generated pseudo-labels alone are sufficient to generate superior results as compared to a model trained only on LIBRISPEECH. The model trained on LIBRISPEECH and LIBRIVOX pseudo-labels achieves an improved 2.28% and 4.99% on dev-clean and dev-other, respectively.

## D. End-to-End Acoustic Models Learn a Language Model: Removing the LM from ASR

In the sections that follow, we show two results. We first give a simple experimental framework to demonstrate that acoustic models trained on speech learn nontrivial language models, and that training on additional audio facilitates learning better acoustic representations. We then show that with a large collection of pseudo-labeled audio, well-trained acoustic models no longer benefit much from decoding with an external language model in most cases.

### D.1. AMs learning LM: transcribing shuffled audio

The language modeling properties of end-to-end acoustic models are briefly discussed in (Chan et al., 2016), where

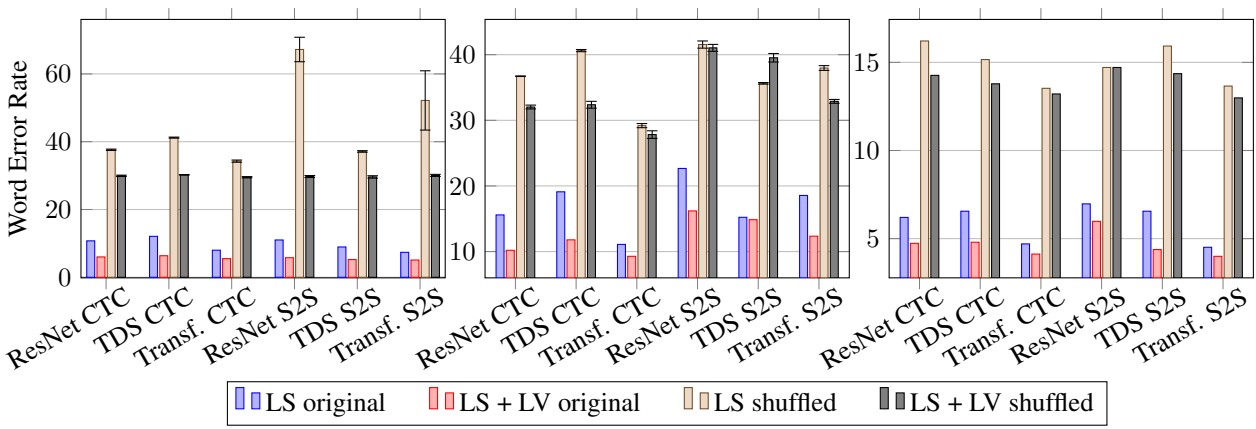

Figure 2. `dev-other` WERs without decoding across acoustic models and loss functions for original and shuffled versions of `dev-other` across three settings. Each plot uses the following original and shuffled audio: *Left*: original and shuffled `dev-other` audio segmented using ASG. *Middle*: audio generated by TTS vocoder for the original and shuffled transcriptions from `dev-other`. *Right*: original and shuffled audio for a subset of `dev-other` recorded by the paper's authors.

an AM trained with CTC is shown to learn an implicit language model based on its predicted posteriors for words with multiple spelling variants. Still other results show that fusing an LM with an AM during training can improve performance (Sriram et al., 2017; Chorowski & Jaitly, 2016; Wu et al., 2016). These previous works use RNN-based acoustic models, which possess infinite receptive fields and processes most or all of an input utterance during a single forward pass. We show that modern convolutional architectures have large receptive fields and likely also learn word representations directly from audio.

If an AM learns a robust LM, the acoustic model will less effectively predict utterances of high underlying word-perplexity; the model will rely on its acoustic representations to predict words without context, providing a good proxy for the quality of its learned acoustic representations. In the experiments that follow, we introduce a simple "shuffled transcription" task in which models transcribe LIBRISPEECH `dev-other` with utterances corresponding to both unshuffled and shuffled transcriptions. Experiments are performed in three audio settings to eliminate bias when scrambling words. First, with a TTS model, unshuffled and shuffled sentences are forwarded through a WaveRNN vocoder (Kalchbrenner et al., 2018) trained on the LJSpeech dataset[7] using the Mozilla TTS toolkit[8]. In the second setting, audio is segmented at the word level using a convolutional stride 2 letter-based AM trained with ASG loss (Collobert et al., 2016), then re-spliced together in the given shuffled order. Finally, the paper's authors recorded unshuffled and shuffled utterances from a subset of `dev-other`.

Figure 2 contains the WERs across audio settings on

---
[7] https://keithito.com/LJ-Speech-Dataset/
[8] https://github.com/mozilla/TTS

`dev-other` without decoding. Both CTC and Seq2Seq models perform poorly across the board on shuffled audio which is expected. As soon as we are interested not in the absolute WER values but in the relative WER values across models / losses / datasets, the main outcome from Figure 2 is that AMs trained with LIBRIVOX pseudo-labels are able to learn better acoustic representations which improve performance on shuffled inputs for which their internal LMs is not predictive.

### D.2. With enough unlabeled audio, decoding with an LM doesn't improve performance

The importance of the language model to the success of the pseudo-labeling is known; (Kahn et al., 2019a) show that in the end-to-end setting, as the quality of the language model used to generate the pseudo-label decreases even marginally, the quality of the model trained on the resulting pseudo-labels decreases. In what follows, we show that through the self-training procedure, decoding an acoustic model trained on LIBRIVOX pseudo-labels generated with the help of a language model gives very small improvements compared to models trained only on LIBRISPEECH.

Results are shown in Figure 3. We use a beam-search decoding procedure without an LM ("Zero-LM") to disambiguate the effect of beam search on WER, and evaluate on `dev-other` to provide a better lower bound for how much decoding with the LM can improve performance (decoder parameters are also optimized on `dev-other`). The models for which results are shown are trained on pseudo-labels from LIBRIVOX generated with an $n$-gram language model without an overlapping text corpus (see the ablation in Appendix C and Section 2.2). Decoding with the LM gives little to no gain for models trained on LIBRISPEECH +

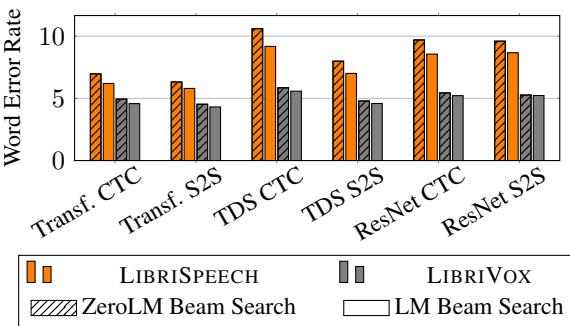

*Figure 3.* WER on `dev-other` for models trained on LIB-RISPEECH and LIBRISPEECH + LIBRIVOX after decoding with and without the 4-gram LM described in Section 2.2. The gain from LM beam-search decoding for models trained on LIB-RIVOX is much smaller compared to that for models trained on LIBRISPEECH.

LIBRIVOX and a much more significant gain for those models trained only on LIBRISPEECH, suggesting information from the 4-gram LM used to generate pseudo-labels on LIB-RIVOX has thoroughly diffused into AMs trained with those labels. Full results can be found in Table 3.

## E. Experiment Details

Comprehensive WER results for LIBRISPEECH and LIB-RIVOX acoustic models, including with greedy and beam-search decoding with different LMs and beam rescoring can be found in Table 3. This section mainly focus on providing details of how we optimize the beam-search decoding and rescoring procedures for our acoustic models.

### E.1. Beam-Search Decoding

When beam-search decoding, we use the `dev-clean` and `dev-other` sets as validation sets and use random search to optimize decoding hyper-parameters. The search ranges of those hyper-parameters are listed in Table 6. We use between 64 and 128 runs in each random search with hyper-parameter values uniformly sampled from the given ranges. It is worth noting that the optimal ranges for language model weight for models trained on LIBRISPEECH are higher than ones found for LIBRIVOX models as shown in Table 7. This is conceivably additional evidence that models trained with additional audio rely less on language models.

### E.2. Rescoring

To perform rescoring, we first dump all hypotheses proposed during beam-search decoding using the optimal hyper-parameters found with random search. When dumping candidates, beam size, token beam size, and beam threshold are increased so as to increase the number of proposed hy-

*Table 6.* Hyper-parameter values and ranges used in a random search for beam-search decoding with $n$-gram (top block) and GCNN (bottom block) LMs.

| | LIBRISPEECH | | LIBRIVOX | |
| PARAMETERS | CTC | S2S | CTC | S2S |
|---|---|---|---|---|
| BEAM | 500 | 50, 100 | 500 | 20, 50, 100 |
| TOKEN BEAM | 100 | 10, 50 | 100 | 3, 5, 10 |
| LM WEIGHT | [0, 3] | [0, 2] | [0, 1.5] | [0, 1] |
| THRESHOLD | 100 | 10, 50 | 100 | 5, 10, 50 |
| WORD INSERT. | [−3, 3] | - | [−3, 3] | - |
| EOS-PENALTY | - | [−10, 0] | - | [−10, 0] |
| BEAM | 250 | 50 | 250 | 20, 50, 100 |
| TOKEN BEAM | 100 | 10, 18 | 100 | 3, 5, 10 |
| LM WEIGHT | [0, 3] | [0, 2] | [0, 1.5] | [0, 0.8] |
| THRESHOLD | 20 | 10, 15 | 20 | 5, 10, 50 |
| WORD INSERT. | [−3, 3] | - | [−3, 3] | - |
| EOS-PENALTY | - | [−10, 0] | - | [−10, 0] |

*Table 7.* Optimal LM weight ranges (based on WER) for beam-search decoding with $n$-gram (top block) and GCNN (bottom block) LMs found via random search.

| | LIBRISPEECH | | LIBRIVOX | |
| DATA | CTC | S2S | CTC | S2S |
|---|---|---|---|---|
| CLEAN | [0.8, 1.4] | [0.6, 1.1] | [0.2, 0.4] | [0.0, 0.2] |
| OTHER | [1.1, 1.9] | [0.6, 1.2] | [0.5, 0.7] | [0.1, 0.5] |
| CLEAN | [0.4, 0.8] | [0.2, 0.5] | [0.2, 0.5] | [0.0, 0.4] |
| OTHER | [0.5, 1.1] | [0.3, 0.7] | [0.3, 0.6] | [0.2, 0.4] |

*Table 8.* Parameters values used when dumping beam candidates for rescoring with $n$-gram (top block) and GCNN (bottom block) LMs.

| PARAMETERS | CTC | S2S |
|---|---|---|
| BEAM | 2500 | 250 |
| TOKEN BEAM | 1500 | 150 |
| THRESHOLD | 5000 | 150 |
| BEAM | 250 | 250 |
| TOKEN BEAM | 100 | 100 |
| THRESHOLD | 20 | 100 |

potheses on which to run rescoring. Further details are listed in Table 8. We find optimal values of rescoring hyper-parameters $\alpha_1$, $\alpha_2$ and $\beta$ (see Appendix B.2) via a grid search for CTC models ($\alpha_1, \beta \in [0, 1]$ and $\alpha_2 \in [−0.3, 0.3]$ where the grid step is set to 0.1), and a random search for sequence-to-sequence models ($\alpha_1, \in [0, 2.5]$, $\alpha_2 \in [−1, 1]$, $\beta \in [−3, 3]$ with 1000 attempts).

## F. Generating Shuffled Audio

This section provides details of how we generated shuffled utterances used in the experiments in Section D.1. Each experiment could introduce systematic error. Therefore, we propose several experiments to conclude. For the two methods generating existing or using new audio (**TTS** and **Segmentation**), we shuffle `dev-other` five times and report the mean and standard deviation (as error bars) in Figure 2.

### F.1. TTS

For each sentence in `dev-other`, we randomly shuffle its words to form a new sentence. We run the resulting text through a TTS model as outlined in Section C to create synthetic audio for the scrambled sentences. While simple and easy to implement, this method introduces and amplifies intrinsic errors in the TTS model into the ablation. In particular, the model struggles to handle many of the rare words present in `dev-other`. Also TTS approach is still away from the human speech.

### F.2. Segmentation

With this method, we first force-align the transcriptions of `dev-other` to the existing audio using a letter-based stride-two ASG model as outlined in Section C and collecting the beginning timestamp and duration of each word. Then, to avoid splicing words that are ready closely together, audio samples are only split when silence of longer than 130 milliseconds is detected (split is done in the middle of silence segment). Finally, audio chunks are randomly shuffled and re-assembled into new utterances. Since this ablation aims to remove LM-friendly context from audio, we filter the resulting recombined audio samples. In particular, we filter all utterances that have only one segment, or have at least one segment with more than 6 words in it. After filtering, 1969 out of 2864 samples in `dev-other` remain. The distribution of the number of words in each of the resulting segments is shown in Figure 4.

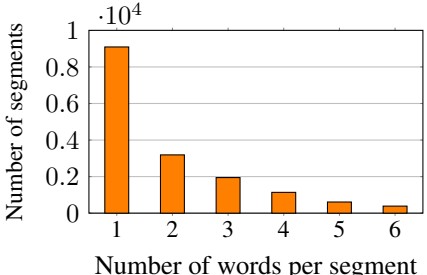

*Figure 4.* Distribution of all $n$-grams in the obtained segments of filtered `dev-other` (1969 samples with 16,362 segments in total).

Unlike the TTS method described above, the segmentation

method reuses audio as much as possible from `dev-other`. That said, neither the force alignment nor the segmentation techniques handle all the word boundaries. As such, there may be incomplete words in the resulting audio and LM-friendly context.

### F.3. Recording

The paper's authors recorded 184 randomly selected sentences from `dev-other` as well as a single set of shuffled utterances. The unshuffled recorded audio has the lowest WER among all the three methods. We plan to complete a collection of unshuffled and shuffled audio for `dev-other` in future work.

### F.4. Perplexity

As shown in Table 9, there are large gaps between the perplexity of transcriptions in the original and shuffled sets across all settings. Our shuffling strategy thus removes important word context and breaks the alignment of the audio words distribution with the LM. The WER gap between the two sets is thus a proxy for the amount of language modeling an acoustic model may implicitly perform.

*Table 9.* Performance of word-level 4-gram and Transformer LMs from Table 1 on original and shuffled audio transcriptions generated from LIBRISPEECH `dev-other`.

| SETTING | SHUFFLED | 4-GRAM LM | TRANSF. LM |
|---------|----------|-----------|------------|
| TTS | NO | 147 | 50 |
| TTS | YES | $749 \pm 2$ | $389 \pm 2$ |
| SEGMENT. | NO | 167 | 56 |
| SEGMENT. | YES | $827 \pm 5$ | $743 \pm 9$ |
| RECORDING | NO | 162 | 49 |
| RECORDING | YES | 3807 | 2995 |