# OpenReview forum: "End-to-End ASR: from Supervised to Semi-Supervised Learning with Modern Architectures"
_ICML.cc/2020/Workshop/SAS — SAS 2020_

### Official Review · AnonReviewer2 · 2020-06-26
**Well-written paper with competently designed experiments and informative results, but some discussions are missing**

**Rating:** 8
**Confidence:** 5

**Review:**

The manuscript presents state-of-the-art results on LibriSpeech with end-to-end speech recognition models trained on additional pseudo-labeled data from LibriVox. The authors show that semi-supervision is key to bridging much of the performance gaps between model architectures, training regimens, and even rescoring strategies. This further underscores the viability of end-to-end ASR going forward, without necessarily relying on external LMs.

Pros

* this avenue of investigation is interesting and the choice of LibriVox as a source of semi-supervised data makes sense.
* the experiments are competently designed and executed, and include a comprehensive cross-section of model architectures.
* the paper is well written and the methodology well described.
* the results are informative to a large fraction of the audience.

Cons

* to me, perhaps the most salient observation of the paper is the excellent level of performance achieved when training on pseudo-labels only - why not give it more prominent mention, perhaps even including it in Table 2?
* another important observation is that information from the 4-gram LM used to generate pseudo-labels on LibriVox has thoroughly diffused into AMs trained with those labels - this calls into question whether a more powerful LM might have produced an even better outcome
* in the same vein, and more generally, there is a critical discussion missing on how pseudo-label quality might influence the conclusions of the paper - as written, the paper remains silent regarding potentially deleterious effects of wrongly labeled data.

---

> ### Author Response · Authors · 2020-07-07
> **Response to Reviewer**
>
> Thanks for your time and helpful comments!
>
> *To address the feedback listed in the cons, respectively:*
>
> - There is unfortunately not much space with which to include those results given the page limit and our other analysis; we agree that those results deserve more attention nonetheless.
>
> - We are continuing to work through these experiments orthogonally; we observe that using a better LM for pseudo-label generation provides better performance.
>
> - We recently completed a [follow up study](https://arxiv.org/abs/2005.09267) on Iterative Pseudo-Labeling where we demonstrate that we can effectively iterate on pseudo-labels generated from a weaker model.

---

### Official Review · AnonReviewer1 · 2020-06-29
**Simple semi-supervised training with advanced models**

**Rating:** 8
**Confidence:** 4

**Review:**

Pros
+ Extensive experimentation
+ Extensive analysis
+ State-of-the-art results at the time of the Arxiv submission.
+ Use of an open source dataset available to anyone

Cons
- Supervised results have since been surpassed by much smaller models.
- The semi-supervised results have been surpassed by supervised models.
- Models are very large and require compute beyond what is available outside large industry labs.
- Given the results are on Librispeech, it is unknown how this would translate to real data.
- The only approach considered is the basic approach that has been used for 30 years.

This paper sets a new bar on a dataset that has become the standard for measuring progress in Seq2Seq modeling. The paper compares multiple model architectures and objective functions. The paper is also an engineering achievement as the authors put together massive amounts of data, large acoustic and language models, and extraordinary compute to push the state-of-the-art on the Librispeech dataset. While the techniques themselves are not novel, there is little work in the literature applying semi-supervised training to Seq2Seq models. The analysis and conclusions are novel. Based on this, this paper should clearly be accepted to the workshop. Of course I do have some criticisms and comments listed below.

1) I'm not sure exactly how the claims about having the "best" result should be handled. It was true when this paper appeared on the Arxiv, but ContextNet and Conformer appeared in May, about a month before the submission deadline for this workshop. They won't appear in peer-reviewed venue until October at Interspeech. I think there should be a footnote mentioning the results have been surpassed since the original submission.

2) In the abstract, you state "...propose several ways
of evaluating the characteristics of unlabeled audio which improve acoustic modeling...". I'm not seeing this. Is the shuffling of the test set what you mean here? To me this means you have ways of evaluating the untranscribed data used for SST, but I don't see it mentioned.

3) Figure 1 is a little confusing. What is the purpose of connecting the points in the plot? I don't think there is a relationship between the models used that confers some kind of ordered relationship. Maybe if the models differed only in the number of parameters, this would make more sense. There must be a better way of representing the points to deliver your message.

4) All of the results use two digits beyond the decimal point. Given the size of the datasets, I think one digit is reasonable. Differences at the second digit are certainly not significant. I also find it makes it difficult to scan the tables and quickly see the magnitude of the differences in results.

5) Footnote 3: "htwtps" --> "https"

6) There seems to be significant care in making sure the initial LibriVox decode is not polluted with either speakers from the original Librispeech dataset or text from the books. I also like that you explored whether this care had much of an effect on the overall result.

7) I like Figure 3 and the conclusion that the benefit from an external LM drops as the acoustic model is trained on more data is interesting. I'm surprised that the effect is the same both for the CTC and Seq2Seq models. I guess since the context window for the CTC model is so large it can still use this high level information even though it does not have access to the predictions the model is making at previous frames.

8) I think the shuffling experiments were a clever approach to disentangling the LM aspect from the AM aspect of the model. It is interesting that the SST seems to provide little benefit when the data is produced by TTS or recorded by the paper's authors after shuffling. In Figure 2, why isn't the pattern consistent? Assuming SST significantly improves the AM aspect of the model, then you still see a large gain when the data is shuffled. When using the TTS or new recordings, there is an acoustic mismatch, so SST does not help on the acoustic side. There is still a gain on the unshuffled data because the model has also learned a better language model. When shuffling the data, the extra LM knowledge is no longer helpful and neither is the acoustic knowledge because of the mismatch. Is that what is happening here?

9) Based on the paper, it looks like the entire Librivox dataset was decoded once with a model that is much worse than your final result. Do you have any sense how sensitive your models are to the initial WER of the transcription model? Given that the final model is so much better, do you get a further gain from iterating the process again using the best SST model for transcription?

10) Have you tried a similar approach on any internal datasets and were results similar? There is extensive literature (mostly with hybrid models) dealing with things like data selection and weighting. Is this something you have explored? You show a consistent gain from adding more data. Instead of randomly selecting data, would performance improve at lower levels (e.g. 1k hours) by ordering the selection by confidence?

---

> ### Author Response · Authors · 2020-07-07
> **Response to Reviewer**
>
> Thanks for your time and helpful comments!
>
>  > “The only approach considered is the basic approach that has been used for 30 years.”
>
> We agree with this assessment — in this work, we aimed to push the limits of this known simple technique in speech recognition in a setting where a large amount of unsupervised data from the same domain is available.
>
> *Regarding specific feedback points:*
>
> 2) Yes, we are referring to the shuffling experiments and the three techniques for conducting them. Given these experiments, we posit that AM performance improves because of better learned acoustic and linguistic representations.
>
> 3) Connecting the dots serves no purpose indeed — we appreciate the feedback, agree, and will remove this for the final version of the paper.
>
> 4) We use two digits when showing results to better-anticipate questions and remove ambiguity around rounding when comparing improvements of 0.1. For Librispeech in particular, we have encountered and asked these questions before.
> 5) Will correct. Thanks.
>
> 7) Yes, we still have a large-enough receptive field to potentially capture a 4-gram LM (especially since the model has stride of 8 or 16). CTC models can train with a large amount of this data, perhaps because they are alignment rather than LM focused.
>
> 8) Yes, SST provides smaller improvements for TTS and our recordings as compared to segmentation experiments because of the number of out of domain speakers and the way the audio was produced. In the segmentation experiments, we use original LibriSpeech audio, so there is an additional systematic error added to TTS and our custom recording methods. We are not comparing the quality of these experiments outright, but are looking at the overall trend and that for each setup (each setup has its own systematic error; we aim to show that there is a trend outside of this error). Even in the TTS and custom recording setups, SST still improves performance on shuffled data; here, systematic errors may have a larger effect. There is indeed still a gain on the unshuffled data because the models have learned better language models.
>
> 9) We have performed additional experiments and published them in an [additional recent work](https://arxiv.org/abs/2005.09267); see Table 3 and the “LS-960 LV-54” results. With more rounds of PL, performance can be improved further. We also show how to learn from a weaker model and how to perform multiple pseudo-labeling rounds.
>
> 10) Indeed, we did not explore weighting and data selection here; we aim to show that the simplest possible self-training approach works.

---

### Official Review · AnonReviewer3 · 2020-06-29
**Interesting paper with practical value**

**Confidence:** 5
**Rating:** 7

**Review:**

This work published on Arxiv few months back has traction in ASR research community for achieving the state-of-the-art on Librispeech then. The paper argues in favour of semi-supervised learning through strong experimental evidence on three prominent acoustic modeling architectures.
While the paper presents no novel ideas, it documents the useful information and hyper parameter settings for large scale ASR training with 1000s of hours of training data. The results from ablation studies highlight some interesting insights.
In my opinion, this paper has significant practical value for researchers in ASR community and is suitable for publication in this workshop.

---

> ### Author Response · Authors · 2020-07-07
> **Response to Reviewer**
>
> Thanks for your time and helpful comments!

---

### Decision · Program_Chairs · 2020-07-01

**Decision:**

Accept

**Comment:**

Dear author(s),

Thank you very much for your submission at the ICML2020@SaS workshop (https://icml-sas.gitlab.io/). Based on the scores assigned by the reviewers, we are happy to notify you that your paper was accepted for the workshop.

Please, address the comments of the reviewers and submit the camera-ready version by July 8. We ask the authors to record a 15min video for your talk. At the workshop, we will have the pre-recorded video as well as a live QA session. It is important to keep this time limit, otherwise, your talk will be automatically cut. The deadline for uploading the video is July 8. The detailed instructions for uploading will follow.

Feel free to contact us for any questions!

Best,

The ICML20@SaS organizers:
Mirco Ravanelli
Titouan Parcollet
Dmitriy Serdyuk
Devon Hjelm
Bhuvana Ramabhadran